# An Image Processing Approach for Real-Time Safety Assessment of Autonomous Drone Delivery

**Assem Alsawy** [1,2,*], **Dan Moss** [3], **Alan Hicks** [3] and **Susan McKeever** [1]

1. School of Computer Science, Technological University Dublin (TU Dublin), Grangegorman Campus, D07 ADY7 Dublin, Ireland; susan.mckeever@tudublin.ie
2. Faculty of Computer Science and Information Technology, Ahram Canadian University (ACU), 6 October City, Giza 12451, Egypt
3. Manna Drone Delivery, Nexus Ucd, D04 V2N9 Dublin, Ireland; dan.moss@manna.aero (D.M.); alan@manna.aero (A.H.)
* Correspondence: assem.abdelhak@tudublin.ie

**Abstract:** The aim of producing self-driving drones has driven many researchers to automate various drone driving functions, such as take-off, navigation, and landing. However, despite the emergence of delivery as one of the most important uses of autonomous drones, there is still no automatic way to verify the safety of the delivery stage. One of the primary steps in the delivery operation is to ensure that the dropping zone is a safe area on arrival and during the dropping process. This paper proposes an image-processing-based classification approach for the delivery drone dropping process at a predefined destination. It employs live streaming via a single onboard camera and Global Positioning System (GPS) information. A two-stage processing procedure is proposed based on image segmentation and classification. Relevant parameters such as camera parameters, light parameters, dropping zone dimensions, and drone height from the ground are taken into account in the classification. The experimental results indicate that the proposed approach provides a fast method with reliable accuracy based on low-order calculations.

**Keywords:** unmanned aerial vehicles; UAV; autonomous drone; drone delivery; image processing; segmentation; safety assessment classifier

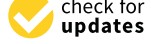



## 1. Introduction

Over the last two decades, drones have become more popular and widely used in a variety of industries [1]. Many fields are now using drones, such as surveying and mapping, security, construction, videos, forestry and agriculture, energy and utilities, goods delivery, public safety and emergency, infrastructure and transport, and mining [2,3]. The utility of drones for capturing aerial information has risen steadily in their roles as machines that can fly and be remotely controlled [4]. Adding intelligence to this machine enables it to function autonomously, effectively creating a flying robot. With the use of artificial intelligence capabilities, autonomous drones transform into intelligent machines capable of performing tasks that typically require human intelligence [5]. However, drones are becoming susceptible to a variety of security threats, such as spoofing, jamming, and information theft. The appropriate design of strong security protocols is essential to counter such attacks and security threats. Several studies have been conducted in this area to fill the gaps in the knowledge, especially concerning secure communication between unmanned aerial vehicles (UAVs), support for perfect forward secrecy, and non-repudiation. In 2021, a security protocol was proposed by Ko, Yongho, et al. [6] consisting of two sub-protocols to safeguard communication between UAVs, and between a UAV and ground control. In 2022, Krichenet et al. [7] discussed drone communication security challenges such as potential risks, attacks, and defenses. Adopting formal methods based on the usage of complex

mathematical models in order to achieve rigorous results will be an intriguing strategy for future efforts linked to drone security [8].

Industrial applications that can utilize autonomous drones have risen over the past ten years with the advances of mobile embedded computing that enable the integration of sensors and controllers into drone platforms. The development of drone intelligence and decision making based on learned processing of sensor data from the internal and external environment is now a part of drone manufacturing [9].

Product delivery, with faster and safer journey times in built-up or difficult-to-reach areas, is a rapidly emerging use of drones. Delivery products include food, medical supplies, and books [10] but could stretch to any products that fall within the size, weight, and robustness requirements for carrying by drone. When supported by manual intervention, each drone typically involves at least two people per flight: one person to pilot or control the drone and another to ensure the safety of the drone during navigation and product-dropping operations. This includes ensuring that the drone will not be stolen and the delivery package is delivered safely to the right place [3]. For viable large-scale growth, drone delivery will require more automation to reduce the dependence on and cost of manual intervention.

### 1.1. Challenges Facing Autonomous Drones

Current drones already contain a high degree of autonomous functions and are effectively almost self-piloting [11]. Drones can perform multiple flying tasks such as destination finding and point-to-point flying without a human at the controls. However, there are still gaps in automation that require the intervention of human decision making. The majority of research efforts in drone automation are now concentrated around the more complex decision tasks of flight planning and navigation, such as enabling the drone to generate its trajectory using Trajectory Planning Algorithms (TPA) [12]. The take-off process is another challenging part of any flight, whereby the drone needs to take off while avoiding houses, pets, people, clotheslines, and other urban area obstacles. While much progress has been achieved in self-navigation, obstacle avoidance during navigation is still a challenging function to fully automate. The obstacle can be a predictable and fixed object such as a tree or a building or it can be a moving object such as a bird, an airplane, or even another drone [13]. There are also some studies focused on the automation of drone landing, on both fixed land and moving platforms (such as a ship) [14]. Currently, a robust, reliable, and accurate self-controlled landing capability is not available [15].

For the delivery aspect of drone delivery, considerations include the selection and/or assessment of the target dropping space as a safe space to deliver a package. Safety assessment should continue for the duration of the dropping process so that unexpected interruptions such as vehicles entering or package grabbing at the point of delivery can be detected. When the delivery drone arrives at the target zone, a classification process is required to classify the dropping zone as a safe or unsafe area, considering whether there is enough obstacle-free space to drop its load. The drone then hovers over the delivery area while the drone lowers the wire to the ground and releases the connected package. For the duration of the delivery, it is necessary to know if there is any change in the delivery zone and whether it should trigger an abort of the delivery process.

### 1.2. Image-Based Approaches

This work attempts to find a solution to the problem of evaluating safe delivery areas for drones, allowing for the context of limited processing resources on a drone and the level of sophistication of the task at hand. Camera images from the drone provide the information-rich input needed for the assessment of the delivery environment. Within the scope of computer vision (CV), image processing concerns the manipulation and analysis of images through the tuning of various factors and features of the images [16]. Transformations are applied to an input image, and the resulting output image is returned. There are many purposes for the processing of images including as a preprocessing step for

image classification, or image segmentation. It can also be used as a standalone approach for these tasks when a learning-based approach is not much needed.

Image classification is the process that classifies an image by assigning it to a specific label. Classifying and naming groups of pixels within an image according to specific rules is also an aspect of image classification [17].

Image segmentation, such as cluster-based segmentation and binary segmentation, typically creates a pixel-by-pixel mask for each object in the image, providing detailed information about the size and shape of image contents. Clustering or cluster-based segmentation is useful for finding the similarity within an image, and then grouping the image pixels into a certain number of groups to be more meaningful and easier to analyze. The typical use of clustering is to categorize the image content [18]. Binary image segmentation is the simplest form of image segmentation. The entire image is split using only two colors to represent two categories. The typical use of the binary method is to separate foreground objects of an image from pixels belonging to the background [18].

Image processing presents simple fast solutions for many computer vision problems, allowing for limitations or assumptions in the problem specifications. These solutions can be sufficiently reliable and more suitable in some scenarios than complex learning model solutions—which typically require heavy computation and high hardware specifications and are thus challenging for lightweight mobile drone devices.

*1.3. Motivation*

Currently, there is no automated solution for a drone-based assessment and overseeing of a safe drone delivery process. Automated delivery requires a fast warning if the selected dropping zone is classified as unsafe for dropping. In addition, tracking the dropping zone during the dropping process to ensure ongoing safety is also critical. At present, there is an assumption in drone automation studies that a safe dropping area is a safe landing area, such that no separate investigation for automation of the dropping process is conducted. However, there are fundamental differences between drone landing versus dropping processes, particularly around the requirement of assessing the suitability of the zone surface. Moreover, research that investigated the approaches to assessing landing zones worked on still images. These are inadequate for dropping zone assessment, which has to take into account the changes that may occur during dropping—such as interventions of an object such as a human or an animal that may affect the safety of the dropping process.

*1.4. Overall Aim*

Solving the problem of ensuring safe package dropping/delivery is a significant step towards automating drone delivery. This work aims to produce a first safety evaluation of the dropping zone immediately before the dropping operation, detect critical changes, and assess the safety of the dropping zone during the entire dropping process. The contributions of this work are twofold: firstly, to study the similarities and differences in the landing and delivery dropping processes; secondly, to build a simple real-time safety classifier for dropping zones using an image segmentation method combined with a motion detector. This paper is structured as follows: The current research approaches for drone landing automation are reviewed in Section 2. Section 3 demonstrates the proposed approach for evaluating the safety of drone delivery dropping zones and describes the evaluation metrics. Experimental tests are presented, and the results are discussed in Section 4. Finally, Section 5 concludes the proposed approach and states the future work.

**2. Related Work**

In 2022, Alsawy et al. proposed an image-based classifier to evaluate the safety of a dropping zone utilizing a single onboard camera. Experiments conducted on a video frames dataset demonstrated that the classifier achieved a precision and recall of 97% in specific test scenarios [19]. To the best of our knowledge, there is no other published research that specifically addresses safety dropping for drone delivery. However, given

the overlaps between this problem and autonomous landing for drones, reviewing the progress on research for safe landing detection will go some way to understanding the extent of work carried out to date.

Analyzing the drone landing literature, we note that outdoor landing of autonomous drones can be categorized into two categories, *normal landing* and *emergency landing*. In a normal landing, the landing would be on a predefined landing zone, and the drone can recognize a safe landing zone using either visual or non-visual information sources. The visual mechanism relies on training the drone to recognize certain known land markings. Landing sites can be identified visually by marking them with certain shapes such as squares, circles, and H-shapes, typically using specific unique colors. The landing zone can also be marked by a "quick response" code (QR code). In the non-visual mechanism, the landing can be in an open area such as a known wide green field. In such cases, GPS technology is commonly used by companies to control a normal landing. Some research approaches supplement GPS information with the use of onboard sensors to more accurately detect a known landing zone. On the other hand, the landing area in an emergency landing is usually unknown; therefore, a suitable landing spot has to be evaluated in real time. The area should have a suitable landing surface and be free of obstacles. The emergency landing methods are largely based on visual mechanisms. Figure 1 presents a categorization of the various approaches to autonomous landing.

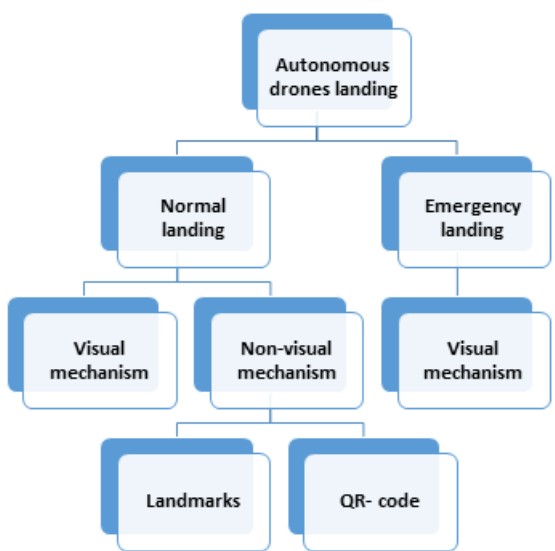

**Figure 1.** Classification of outdoor autonomous drone landing.

The following subsections explore the most recent work accomplished to date in each category.

### 2.1. Normal Landing Approaches

Autonomous landing continues to attract the attention of many researchers. Most methods use GPS information for drone navigation and landing in normal situations as a non-visual mechanism. In 2018, Kan et al. [20] proposed a method to enable the drone to autonomously navigate, hover, and land using rectified GPS information. The authors introduced a calibration of the GPS method to overcome the deviations that have been found with GPS sensors. In their work, the experimental drone was able to autonomously fly to the goal position. In 2019, Patrik et al. used GNSS (Global Navigation Satellite System) and an onboard compass in drone navigation and landing to reduce the positional deviation of the landing position between the initial landing site and the optimum landing site. The experiments showed that the average deviation from the designated mark is less than one meter [21].

Regarding visual mechanisms for a normal landing, in 2017, Huang et al. [22] applied a histogram of oriented gradients (HOG) and linear support vector machine algorithm (SVM) to discover and identify known landmarks in a certain region, achieving an overall accuracy of 84.0%. Moreover, in 2017, Nguyen et al. introduced a safe drone landing method in the absence of GPS signals using the marker-based tracking algorithm based on the visible-light camera sensor. This approach used a unique marker as a tracking target through the landing process with an average error of 0.095% [23]. In 2020, Lee et al. presented a method for detecting specific landing positions for an autonomous landing. This method used an airborne vision system with a fully convolutional neural network to recognize landing markers. The qualitative results show there were some unsuccessful outcomes, which were mainly due to imprecise motion control and lag in the communication and processing [24]. In 2020, Minhua et al. suggested a QR code as an augmented reality tag (ARtag), which was used as the target marker to ensure that the drone could accurately identify the landing area [25]. In 2021, Nio et al. proposed an autonomous drone landing scheme to land on a mobile unmanned ground vehicle (UGV) with high accuracy by utilizing multi-scale QR codes for different heights. A vision-based independent landing algorithm was proposed for both outdoor and GPS-denied situations. Over 85% of the attempts made with this strategy were successful [26].

### 2.2. Emergency Landing Approaches

The problem of autonomous drone emergency landing has also received attention from researchers recently, with all approaches relying on visual input, as shown in Figure 1. In 2019, Loureiro et al. [27] introduced a system that efficiently detects safe landing zones for drones. The main feature of this system is a geometric approach to determine the suitability of candidate landing areas using light detection and ranging (LIDAR) data. The detection algorithm had an average execution time of 37.87 ms. Bektash et al. [28] presented a system that tracks safe places to land, or even to crash, in the event of a failure situation, with potential landing sites selected using a convolutional neural network. The framework has the potential to overcome uncertainties in operations with a safe profile and increased acceptance. In 2021, Gonzalez et al. proposed a lightweight safe landing zone detection algorithm to prevent collisions with people in emergency landing situations. The algorithm is based on a lightweight convolutional neural network framework developed for crowd detection and counting. Real-time experiments validated the proposed strategy via implementation in a low-cost mobile phone application and a commercial drone [29]. Additionally, in 2021, Loureiro et al. [30] investigated the steps of detecting safe landing zones by developing an algorithm that classifies the (LIDAR) data and stores the positions of the most suitable zones. The geometric characteristics are used to discover the possible correct points, with Principal Component Analysis (PCA) then used to find the highest potential zone. The algorithm is tested in simulated scenarios and an experimental dataset, suggesting its suitability to be applied in real-time operations.

Research on landing in normal situations has addressed some issues such as navigating to the landing area using GPS information and recognizing the landing area by detecting landing markers or QR codes. These various normal landing methods are interesting as a mechanism for choosing and finding a predetermined landing point. However, scalable delivery by drone involves many ad hoc destinations, so a mechanism that does not rely on pre-marking is needed. Research on emergency landing has also addressed issues such as evaluating the landing area to find the most suitable area for landing and using LIDAR data to prevent collisions with people. However, the delivery dropping operation requires a continuous assessment of the area of interest for the duration of a dropping operation. These gaps have not been addressed by any of the landing categories.

### 3. The Proposed Safety Assessment Classifier

Drone delivery issues are well recognized by the companies that are already operating in the delivery market. The industry view of any of these issues is a key factor in the

development of autonomous drone delivery technologies, where cost, customer trust, and adherence to regulations are all considered. In a typical delivery transaction, delivery companies require their customers to provide them with GPS coordinates as a target point for exactly where they would like their packages to be dropped. This target point enables the drone navigation system to plan its own course to it. Target points for a product drop may vary in nature. For example, it could be a front yard, a back yard, or even a basketball court. Regardless, it has been selected by the customer as a suitable area in which to place the delivered package. The company's concern is to ensure that this point is safe at the actual time of dropping, i.e., the dropping zone is empty and ready to receive the delivered package, without obstruction, e.g., no human, dog, box, or even tree is in the way of the package. In addition, when the drone arrives at the target point and after the initial verification of safety, the zone must remain safe during the whole dropping process.

### 3.1. Proposed Solution

GPS information helps the drone navigate to the target point or dropping location. Two tasks need to be carried out for a successful dropping at a predesignated delivery spot: the first task is performed on the static image taken *before* the dropping operation starts to decide whether the dropping zone is safe for dropping at this moment or not; the second task is performed on the live streaming images of the dropping zone *during* the dropping to ensure that the dropping zone remains safe and that the safety state has not changed.

#### 3.1.1. Before the Dropping Process: Is the Drop Zone Safe?

The onboard camera mounted on the drone is directed vertically downward to take pictures. When the drone reaches the target point using the GPS coordinates, the camera captures an image of the area directly below the drone. This image includes the image of the dropping zone. To assess the safety of the dropping zone, an image preprocessing and a two-phase approach (segmentation and threshold function) are applied to this image.

#### Preprocessing

Image preprocessing operations are used to clean up, remove any distortion that may exist in the image, and convert the image to that most suited to the analysis task at hand [31]. The sequence of operations applied is shown in Figure 2, and is explained as follows:

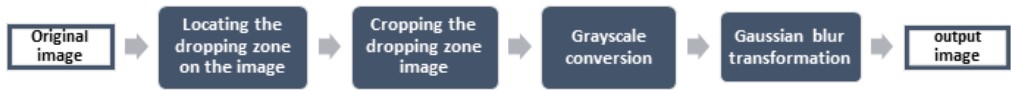

**Figure 2.** The sequence of preprocessing operations.

1.  Locating the dropping zone on the image
    The dropping area should be defined and cropped on the image to match the agreed size of the dropping area so that this cropped image represents the dropping zone. Working with a live camera introduces additional complexities concerning the camera characteristics and lighting factors. In addition, there are certain values related to the dropping preferences of the drone company that need to be determined beforehand. While GPS information helps to locate the center of the dropping zone, it is not enough to find the side length of the box that represents the dropping zone on the image. Camera parameters and dropping parameters must be known in advance. Camera parameters are the parameters that relate to camera characteristics, which are:
    (a)  Focal length (F): the distance in millimeters between the lens of the camera and the camera sensor [32].
    (b)  Sensor size (Z): the size of the camera film that receives the light coming through the lens into the camera, and then converts that into an image [32].

(c)    Camera resolution (E): the number of recorded pixels or the size of the image taken by the camera [33].

Dropping parameters are the hyper-parameters that represent the delivery company's preferences, which are:

1. The altitude (H): represents the height of the drone during the dropping.
2. Zone side length (R): represents the length of the square that marks the dropping zone.

To calculate the square side length of the dropping zone on the image (D) that matches the real zone side length (R), as shown in Figure 3, the two triangles are similar [34]; therefore, the length of the square image on the camera sensor is calculated from Equation (1).

$$D = \frac{R}{H} * F \tag{1}$$

It is necessary to represent this length in a form that is independent of any output device. Therefore, we use Equation (2) to find the number of pixels that represent this length.

$$Q = \frac{D}{Z} * E \tag{2}$$

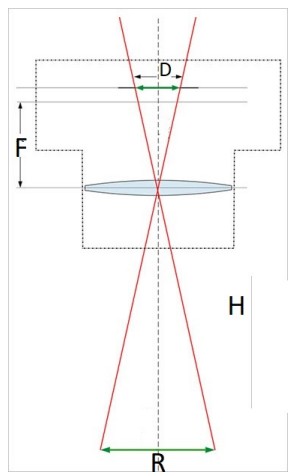

**Figure 3.** Camera and dropping parameters.

1. Cropping the dropping zone image
   When the drone is hovering vertically over the preselected GPS location, the GPS coordinates will be the center of the image ($C_x$, $C_y$), and the dropping zone on the image will cover the ranges $C_x - \frac{Q}{2} : C_x + \frac{Q}{2}$ and $C_y + \frac{Q}{2} : C_y - \frac{Q}{2}$
2. Grayscale conversion
   Grayscale conversion is used to simplify the computational requirements; the value of each pixel in a grayscale image represents only an amount of light [35]. Each pixel with the position $(i, j)$ is converted from the three-channel format, red ($R$), green ($G$), and blue ($B$), into one channel, grayscale ($X_{ij}$), according to Equation (3).

$$X_{ij} = 0.299 * R_{ij} + 0.587 * G_{ij} + 0.114 * B_{ij} \tag{3}$$

3. Gaussian blur transformation
   After obtaining the image in one dimension from the grayscale transformation in the previous step, a blur transformation is applied to each pixel value to reduce image noise and smooth the image according to the Gaussian formula in Equation (4) [36].

$$P_{ij} = G(x_{ij}) = \frac{1}{\sqrt{2\pi\sigma^2}} e^{-\frac{X_{ij}^2}{2\sigma^2}} \tag{4}$$

where $\sigma$ is the standard deviation of the distribution of pixels values.

Figure 4 shows an example of applying preprocessing operations to a drone view image. The original image is shown in (a); (b) represents the image where the dropping zone is located and the calculation of the dropping zone coordinates, with the zone indicated in the yellow box; (c) is the cropped image leaving solely the dropping zone; (d) is the converted image to grayscale; and the final transformation using Gaussian blur on the image of the dropping zone is shown in (e).

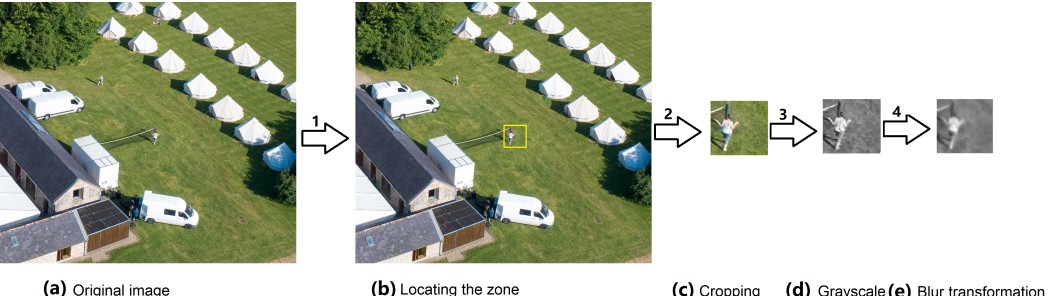

(a) Original image      (b) Locating the zone      (c) Cropping    (d) Grayscale (e) Blur transformation

**Figure 4.** Applying preprocessing operations to a drone view image.

Phase I: Segmentation

The goal of the segmentation process is to change the image representation to a format where it is easier to identify any object from the background. Cluster-based segmentation and binary segmentation are the most commonly used methods and are examined in this approach. The segmentation method is applied to the preprocessing output image, which represents the dropping zone in a suitable format.

The K-means algorithm is used as a clustering method for dividing an image into categories, where K is the chosen number of categories [37]. In binary segmentation, only two colors are used to represent the two categories. Each pixel in the image, having position $(i, j)$ and the value $(P_{ij})$, belongs to one of the two classes. The main task in this method is to determine the threshold value that separates the two categories, and determines which category a pixel is in. This threshold value is sensitive to light. If the camera is operating in broad daylight, the threshold should be low, and it should be high if the images are taken at night. The histogram-based thresholding method is adopted to determine the threshold value $\beta$; each pixel value in the segmented image is calculated as in Equation (5).

$$p_{ij} = \begin{cases} 1 & P_{ij} \geq \beta \\ 0 & \text{otherwise} \end{cases} \tag{5}$$

It is worth noting that the greater the object-free space in the zone, the more one segment dominates the image. This segment represents the unoccupied parts of the zone. In the case that the entire region is empty, the overall image is represented by a single segment.

Phase II: Threshold Function

In this phase, the ratios of the segment classes in the cropped image are calculated in order to assess the overall level of obstruction in the image. These ratios represent the ratio of all the objects that may be present in this dropping zone, and also the ratio of empty space that may be possible for dropping. The ratio of empty space ($I$) can be computed as in Equation (6).

$$I = \frac{\sum_{i=1}^{k} \sum_{j=1}^{l} p_{ij}}{N} \tag{6}$$

where $N$ is the number of pixels in an image, $l$ is the number of rows in an image, $k$ is the number of columns in an image, and $N = l * k$. Moreover, if needed, the ratio of objects in the image can be computed as in Equation (7).

$$O = N - I \tag{7}$$

Hence, a threshold function is applied to the ratio of the empty space ($I$) to classify the dropping zone in the image as safe or unsafe, as shown in Equation (8).

$$Output = \begin{cases} \text{Safe} & I \geq \alpha \\ \text{Unsafe} & \text{otherwise} \end{cases} \tag{8}$$

where $\alpha$ is the minimum acceptable empty space ratio in the dropping region. This parameter is set by the delivery company to specify the empty space needed to accommodate the package, and the $\alpha$ value can be varied according to the preferred dimensions of the dropping zone box and the size of the delivered package.

Figure 5 demonstrates the application of the preprocessing and the proposed two-phase approach to an image of a dropping area: (a) shows the original image; the output image of the preprocessing stage for the selected dropping zone is shown in (b); (c) represents the segmented image of the dropping zone resulting from the first phase; and the output of calculating the ratio of empty space within the dropping zone is presented in (d). If this ratio $\geq \alpha$, the zone is deemed safe for dropping; otherwise, it is not safe.

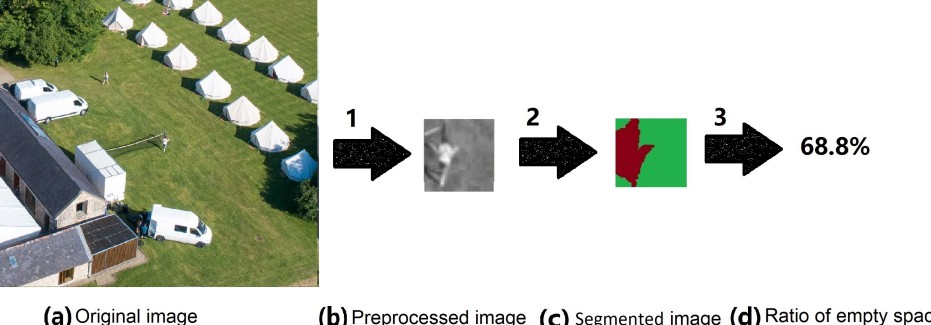

**(a)** Original image     **(b)** Preprocessed image  **(c)** Segmented image **(d)** Ratio of empty space

**Figure 5.** Applying the two-phase approach to an image of a dropping zone, example 1.

Figure 6 demonstrates the application of the two-phase approach to another sample image, with a higher proportion of empty space, and it is more likely to be counted as a safe dropping zone even with a very high safety threshold.

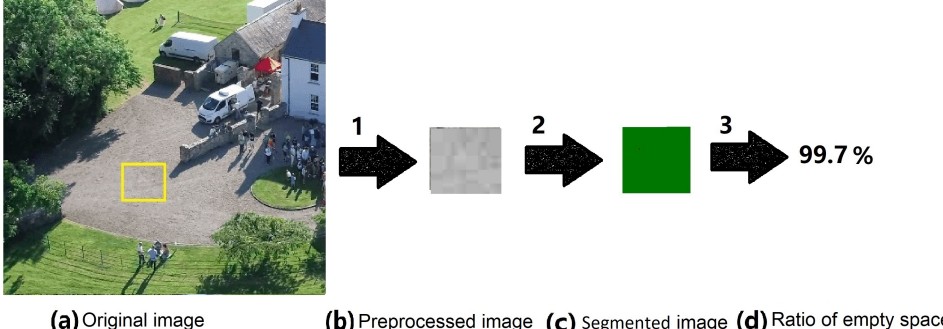

**(a)** Original image     **(b)** Preprocessed image  **(c)** Segmented image **(d)** Ratio of empty space

**Figure 6.** Applying the two-phase approach to an image of a dropping zone, example 2.

3.1.2. During the Dropping Process: Drop Zone Remains Safe?

If the dropping zone is assessed as safe, the drone can begin the process of dropping the delivery package. However, if conditions change during the dropping and any danger

appears in the zone, the dropping may need to be aborted. Therefore, the drone needs to keep monitoring the dropping zone through the live capture of the mounted camera and continue the safety assessment during the entire dropping operation.

In order to further assess safety, GPS information allows us to live-track the dropping area and recalculate its coordinates from one frame to another. The image prepossessing and the two phases should be applied to each frame of the live stream. In the case where the safety assessment result of any frame is unsafe, the drone should hold the package and keep hovering and waiting for the status to change in order to resume dropping as long as the predetermined dropping time has not yet expired. If the dropping time has expired, the drone will cancel the dropping operation.

Figure 7 represents the flowchart of the safe dropping process.

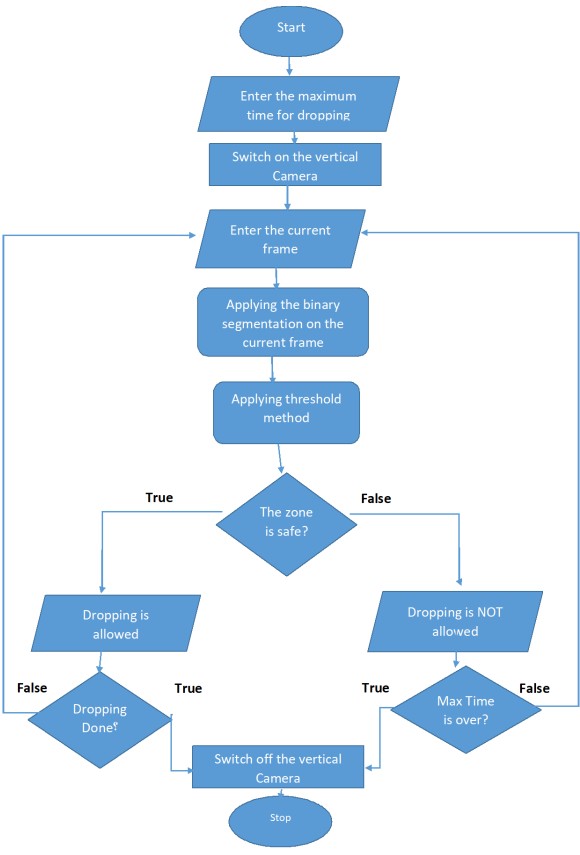

**Figure 7.** Flowchart of the safe dropping process.

### 3.1.3. Improve the Performance: Motion Detection

To maintain the safety assessment of the dropping area during a dropping operation, a loop of functions is performed over and over again, which leads to serious complexity and slow performance. This lag would be an obstacle to an efficient drone delivery process. In order to reduce this complexity, a motion detection function is proposed that checks if any changes have occurred in the dropping region. With that information, and if there is no change from one frame to another, then there is no need to re-execute the assessment steps. The assessment is only implemented if a change is detected.

Since the drone is hovering during the dropping process, motion in the dropping area can be detected by simply detecting changes between every two successive frames. The sum of absolute differences $SAD$ is a suitable measure of the similarity between two images here. It is found when the absolute numerical value of each pixel $P$ in the location $(i, j)$ in an image is subtracted from the corresponding pixel $P'$ in another image, as in the

form $|P_{ij} - P'_{ij}|$. Equation (9) computes the sum of absolute differences $SAD$ to compare two consecutive frames in the live stream.

$$SAD = \sum_{i=1}^{k} \sum_{j=1}^{l} |P_{ij}^m - P_{ij}^{m+1}| \tag{9}$$

where $m$ and $m + 1$ are two consecutive frames numbers, $l$ is the number of rows in each frame, and $k$ is the number of columns in each frame.

In the case of identical frames, $SAD$ equals zero, which indicates that there is no movement, and therefore, there is no change in the safety state. However, a small value $\epsilon$ for the difference between frames may be tolerated due to the mounted camera shaking because the drone is hovering.

Algorithm 1 is the enhanced form of the delivery drone safe-dropping algorithm including the motion detector.

---

**Algorithm 1** Drone Delivery Dropping Safety Assessment Algorithm

---

**Require:** Maximum dropping time in seconds: T, Prepossessed streaming frames: $f_1, f_2, ...... f_T$, the thresholds: $\alpha, \beta, \epsilon$

**Ensure:** Dropping area status is Safe/ Unsafe

---

1:   $t \leftarrow 15 * T$              ▷ One frame is processed every 1/15 second
2:   **while** $t \neq 0$ **do**
3:      $m \leftarrow 15 * T - t + 1$
4:      **if** $m > 1$ **then**
5:         **if** $\sum_{i=1}^{k} \sum_{j=1}^{l} |P_{ij}^m - P_{ij}^{m-1}| < \epsilon$ **then**       ▷ Motion detection
6:           $t \leftarrow t - 1$
7:           *continue*             ▷ Skip to the next frame
8:         **end if**
9:      **end if**
10:     **for** $i \leftarrow 1$ to $k$ **do**            ▷ Binary segmentation
11:        **for** $j \leftarrow 1$ to $l$ **do**
12:           **if** $P_{ij}^m > \beta$ **then**
13:             $p_{ij}^m \leftarrow 1$
14:           **else**
15:             $p_{ij}^m \leftarrow 0$
16:           **end if**
17:        **end for**
18:      **end for**
19:      **if** $(\sum_{i=1}^{k} \sum_{j=1}^{l} p_{ij})/N \geq \alpha$ **then**       ▷ Threshold function
20:        $Output \leftarrow$ Safe
21:      **else**
22:        $Output \leftarrow$ Unsafe
23:      **end if**
24:      $t \leftarrow t - 1$
25: **end while**

---

### 3.2. Evaluation Metric

To evaluate the quality of the proposed image processing classifier, the widely used precision–recall metric is used, precision–recall is a useful measurement for prediction success especially when classes are highly unbalanced [38]. Precision can be viewed as a quality scale that quantifies the number of correct class predictions that actually belong to this class. High precision indicates that the classifier returns more valid results than

incorrect ones. On the other hand, recall can be viewed as a quantity scale that quantifies the number of correct class predictions of all the classes' instances. High recall indicates that a classifier returns the most valid results that exist in the dataset. To provide class-level visibility when performing precision and recall calculations, a confusion matrix as in Figure 8 is used. It has four values as follows:

1.  TS (True Safe): TS represents the number of correctly classified zones that are safe.
2.  FS (False Safe): FS represents the number of zones misclassified as safe but are actually unsafe.
3.  TU (True Unsafe): TU represents the number of zones that are properly classified as unsafe.
4.  FU (False Unsafe): FU represents the number of zones incorrectly classified as unsafe but are actually safe.

| | | Predicted Class | |
|---|---|---|---|
| | | Safe | Unsafe |
| **Actual Class** | Safe | True Safe (TS) | False Unsafe (FU) |
| | Unsafe | False Safe (FS) | True Unsafe (TU) |

**Figure 8.** Confusion matrix.

According to the industry point of view, there is a need to evaluate the performance of the proposed classifier in both safe and unsafe prediction conditions.

Precision (Safe) measures the percentage of zones predicted as safe that are correctly classified, as expressed in Equation (10).

$$Precision(Safe) = \frac{TS}{TS + FS} \tag{10}$$

Recall (Safe) measures the percentage of actual safe zones that are correctly classified, as indicated in Equation (11).

$$Recall(Safe) = \frac{TS}{TS + FU} \tag{11}$$

Precision (Unsafe) measures the percentage of zones predicted as unsafe that are correctly classified, as shown in Equation (12).

$$Precision(Unsafe) = \frac{TU}{TU + FU} \tag{12}$$

Recall (Unsafe) measures the percentage of actual unsafe zones that are correctly classified, as illustrated in Equation (13).

$$Recall(Unsafe) = \frac{TU}{TU + FS} \tag{13}$$

## 4. Evaluation and Discussions

A range of experimental tests were conducted to assess the extent to which the proposed safe dropping zone approach can accurately detect safe versus unsafe zones at the beginning of and during the delivery process. The experiments passed through three stages, with each stage using a different dataset.

### 4.1. Datasets

There are two groups of image data used in this method to enable working on still images and a live camera as follows:

1. The proposed solution aims first to examine the given dropping zone right before the dropping operation and show whether it is safe or unsafe to drop the delivery package. The dataset used to test this classifier consists of static images: 70 bounded boxes were selected as candidate dropping zones and cropped from 14 drone view images. These boxes represent different types of surfaces: 10 boxes of flat horizontal surfaces, 10 boxes of mottled surfaces, 10 boxes of vertical surfaces, 10 boxes of sloped surfaces, 10 boxes of elevated surfaces, 5 boxes of water surfaces, and 15 boxes that contain objects such as cars, humans, trees, and pets. Figure 9 displays samples of the first group of the dataset.

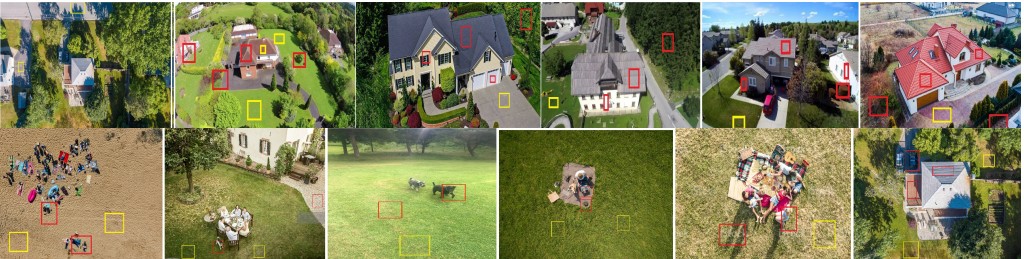

**Figure 9.** Samples from the still images dataset.

2. To track the safety of the dropping zone during the dropping process, 18 recorded-by-drones videos were used, each of which represents the dropping area during the dropping process. Each video has a frame rate of 15 frames/s and are 30 s long, and videos were recorded from different heights (12 m, 14 m, 16 m, 18 m) using different side lengths of dropping area boxes (2 m, 3 m, 4 m, 5 m). Nine videos were for flat horizontal surfaces; the other nine were for surfaces mottled with irregular markings, shades, and colors. To simulate the change in safety situations, various objects were entered into each video at random times for 15 s.

*4.2. Experiments on Static Images*

First, it is required to test the performance of the classifier when assessing the safety of the dropping zone using the still images dataset; these still images represent the dropping area just before the drop operation. Two versions of the classifier were developed using two different segmentation methods, clustering-based segmentation and binary segmentation. Both versions were tested on a still images dataset, then the results were compared to see which one was more suitable for the delivery drop zone classification. In each of these two sets of experiments, we needed to determine the categories of dropping zone surfaces used, the number of images in each category, and the actual status of whether it is safe or unsafe. The categories of surfaces that are primarily considered safe are flat horizontal surfaces such as paved ground, and mottled surfaces such as grass and gravel. Vertical surfaces, sloped surfaces, elevated roofs, and water surfaces are considered unsafe dropping surfaces.

4.2.1. Cluster-Based Segmentation

These experiments aim to use the five-color clustering segmentation method by applying the K-means clustering algorithm (K = 5) to detect five types of segments in the dropping area, and then evaluate the safety state by applying the thresholding function as in Equation (8).

Table 1 shows the experiment results of the clustering-based classifier. The classifier achieved complete accuracy in identifying the unsafeness of the dropping zones that contained objects and also provided high accuracy in identifying the unsafeness of vertical, sloping, and watery surfaces. On the other hand, it failed to recognize the safety of flat horizontal surfaces and mottled surfaces, and the unsafeness of the elevated surface. The reason for this is that the clustering method tried to extract the background of the image, assign one color to it, and assign other colors to different types of objects. Thus, it is suitable

for detecting objects and vertical, sloped, and water surfaces. Flat horizontal surfaces and mottled surfaces are likely to be misclassified by this method, especially if the surface has paint or shadow. Elevated surfaces are recognized as a safe dropping zone with this method, but according to delivery companies, roofs are not suitable places to drop delivered packages. For a better description of the classifier's performance, Table 2 shows the overall confusion matrix.

**Table 1.** Applying a 5-color clustering-based classifier to a still images dataset.

| Cluster Segmentation | | | | |
|---|---|---|---|---|
| Type | No. of Boxes | Correct Output | No. of Correct Results | Accuracy |
| Objects | 15 | Unsafe | 15 | 100.0% |
| Flat horizontal surfaces | 10 | Safe | 6 | 60.0% |
| Mottled surfaces | 10 | Safe | 5 | 50.0% |
| Vertical surfaces | 10 | Unsafe | 9 | 90.0% |
| Sloped surfaces | 10 | Unsafe | 9 | 90.0% |
| Elevated surfaces | 10 | Unsafe | 2 | 20.0% |
| Water | 5 | Unsafe | 4 | 80.0% |

**Table 2.** Confusion matrix (clustering-based classifier).

| | | Predicted Class | |
|---|---|---|---|
| | | **Safe** | **Unsafe** |
| Actual Class | Safe | 11 | 9 |
| | Unsafe | 11 | 39 |

According to these results:

Precision (Safe) = 50%, recall (Safe) = 55%.

Precision (Unsafe) = 81.25%, recall (Unsafe) = 78%.

This method is not reliable for recognizing a safe dropping zone and results in many false alarms saying an area is unsafe when it is safe. However, it achieves a relatively good result when predicting unsafe zones.

4.2.2. Binary Segmentation

This set of experiments applies the binary segmentation method to detect two types of segments in the dropping zone and then predict the safety state by applying the thresholding function.

Table 3 shows the experiments of the binary-segmentation-based classifier. This version of the classifier achieved complete accuracy when identifying the unsafeness of the dropping zones that contain objects and identifying the safety of flat horizontal surfaces, and also provided high accuracy when identifying the safety of mottled surfaces. On the other hand, it failed to recognize the unsafeness of vertical, sloping, elevated, and watery surfaces.

**Table 3.** Applying binary-segmentation-based classifier to still images dataset.

| Binary Segmentation | | | | |
|---|---|---|---|---|
| Type | No. of Boxes | Correct Output | No. of Correct Results | Accuracy |
| Objects | 15 | Unsafe | 15 | 100.0% |
| Flat horizontal surfaces | 10 | Safe | 10 | 100.0% |
| Mottled surfaces | 10 | Safe | 9 | 90.0% |
| Vertical surfaces | 10 | Unsafe | 1 | 10.0% |
| Sloped surfaces | 10 | Unsafe | 3 | 30.0% |
| Elevated surfaces | 10 | Unsafe | 0 | 0.0% |
| Water | 5 | Unsafe | 1 | 20.0% |

Binary segmentation differentiates between the background and the object that may be present in the image. Thus, this classifier is suitable for detecting objects and empty spaces in flat horizontal surfaces and mottled surfaces. Water surfaces according to this classifier are recognized as safe as they are usually detected as a background. Elevated surfaces, vertical surfaces, and sloped surfaces are detected as one-color surfaces, so they are recognized as single-colored surfaces and therefore identified as safe. The confusion matrix in Table 4 summarizes the predicted results of the binary-segmentation-based classifier.

**Table 4.** Confusion matrix (binary-segmentation-based classifier).

| | | Predicted Class | |
|---|---|---|---|
| | | **Safe** | **Unsafe** |
| Actual Class | Safe | 19 | 1 |
| | Unsafe | 30 | 20 |

According to these results:
Precision (Safe) = 38.7%, recall (Safe) = 95%.
Precision (Unsafe) = 95.2%, recall (Unsafe) = 40%.

The classifier shows poor results in terms of precision (Safe) and recall (Unsafe). Recall (unsafe) refers to the percentage of unsafe zones in the dataset that are correctly classified. Given the nature of the task of safety assessment for dropping, it is considered a key factor and real applications require a high recall score.

In reality, however, the situation is not as bad as it seems, and the binary segmentation method may be a more appropriate method to build a classifier if we take into account the real-life ground operations of drone delivery companies. The industry point of view has some assumptions that may give a new perspective to this problem, as explained next.

4.2.3. Assumptions

From an industry point of view, there are two working practice assumptions to consider. These assumptions have emerged from the real experiences of delivery companies, particularly those with delivery drones in operation.

- The first assumption is that the safe landing surface is a safe dropping surface, which means that the standards of the dropping surface are the same as those of the landing surface. The safe landing area is evaluated based on two main factors, slope and roughness, which are the factors by which the surface is classified as a safe landing surface or not [39].

  According to this assumption, a suitable surface for landing and harmless to the drone itself is also suitable for dropping and placing the package on. Knowing that a surface is preselected as suitable for a dropping surface is not a sufficient measure for

dropping. It must also have enough empty space with no obstacles during the time of dropping, whether it is a fixed obstacle such as a box or a toy, or a moving obstacle such as a human or a pet.

- The second assumption differentiates between the landing requirements for both regular and emergency landings and dropping requirements: the dropping area is predetermined by the customer so it will not be a water surface, an elevated surface, or a sloping surface.

In the case of a regular or *normal* landing, which typically uses GPS information to perform a landing either on a controlled area or on a landmark, the drone navigates to the destination point defined by the GPS coordinates, hovers, and then lands or recognizes a landmark to land on [40]. Figure 10 shows the normal landing on the landmark.

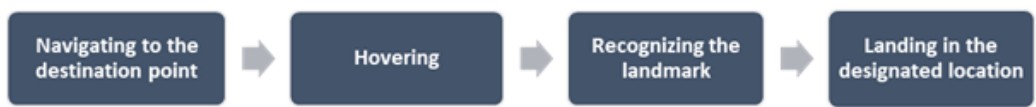

**Figure 10.** Steps of drone's normal landing on landmarks.

In the case of an *emergency* landing, the self-driving drone hovers to assess the safety of the landing area in two steps: the first step is to recognize the type of surface and confirm that it is a solid surface and not an elevated or sloping surface, and the second step is to ensure that there is no obstacle preventing a safe landing [41]. Figure 11 demonstrates the steps for an emergency landing by an autonomous drone.

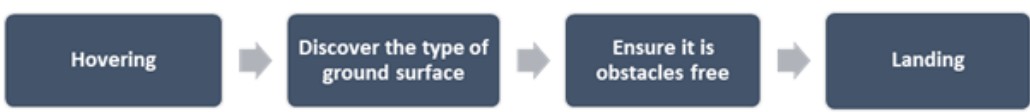

**Figure 11.** Steps of drone's emergency landing.

In the case of a *dropping* process, the drone also navigates to the dropping point using GPS information and hovers to assess the safety of the dropping area. The dropping algorithm is solely concerned with whether the dropping zone is clear of any obstacle either at the beginning of the drop or during the dropping process. Figure 12 shows the steps of safe dropping.



**Figure 12.** Steps of drone's delivery dropping.

Under these assumptions, the only categories that need to be studied are objects, flat horizontal surfaces, and mottled surfaces. From this perspective, there is currently no benefit from studying vertical, sloped, elevated, or water surfaces, and only the first three rows in Table 3 are required. The binary segmentation results using these assumptions indicate improved results when the customer pre-approves the delivery point—and the safety assessment is focused on the safety of a prevalidated space.

### 4.2.4. Comparison of Results

Table 5 displays the values of precision and recall in both the safe and unsafe cases for experiments using the clustering-based classifier, binary segmentation-based classifier, and binary-segmentation-based classifier according to industry assumptions.

**Table 5.** Precision–recall metric for different versions of the classifier.

|  | Precision (Safe) | Recall (Unsafe) | Precision (Safe) | Recall (Unsafe) |
|---|---|---|---|---|
| Clustering | 50.00% | 55.00% | 81.25% | 78.00% |
| Binary segmentation | 38.70% | 95.00% | 95.20% | 40.00% |
| Binary segmentation (under industry assumptions) | 100.00% | 95.00% | 93.75% | 100.00% |

Figure 13 shows a comparison of the performances of the different classifier versions. The binary-segmentation-based classifier under the suggested assumption achieves the highest score on the precision–recall scale.

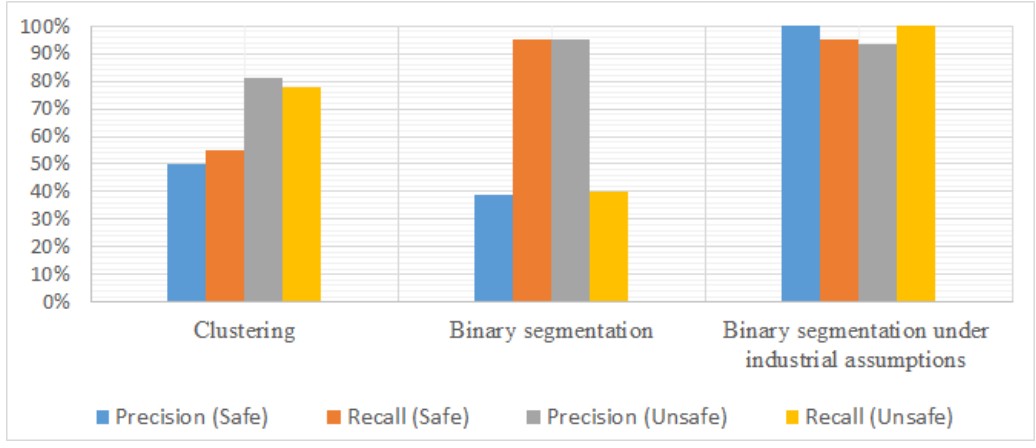

**Figure 13.** Comparing the performance of the proposed classifier versions.

### 4.3. Experiments on Recorded Videos

In addition to selecting a safe placement of the package, the classifier must track the safety of the dropping zone during the dropping process to ensure that there is no change in the dropping zone, and consequently, no change in the safety state from safe to unsafe. A single static image of the dropping area is not enough for classification. Analysis of live camera images is also required.

To test the accuracy of the classifier that recognizes whether the dropping zone is safe or not before and during the dropping process, the second dataset was used, consisting of 18 recorded videos each 30 s long. In each video, motion detection is applied to every two consecutive frames to detect any movement by discovering any change from one frame to the next frame. If no motion is detected, it means that the safety state has not changed. If movement is detected, the classifier analyzes the current frame as a static image by applying preprocessing, the binary segmentation method, and the threshold function to evaluate the safety state in this frame. Table 6 displays the results of applying the proposed classifier to these recorded videos.

**Table 6.** Accuracy of the proposed classifier applied to the recorded videos dataset.

| Type | Actual Output | No. of Frames | No. of Correct Results | Accuracy |
|------|---------------|---------------|------------------------|----------|
| Empty flat horizontal surfaces | Safe | 2025 | 2025 | 100.00% |
| Empty mottled surfaces | Safe | 2025 | 1830 | 90.37% |
| Flat horizontal surfaces with objects | Unsafe | 2025 | 2025 | 100.00% |
| Mottled surfaces with objects | Unsafe | 2025 | 2025 | 100.00% |

The classifier achieved complete accuracy in recognizing the safety of empty flat horizontal surfaces and the unsafeness of the dropping zones containing objects and also provided high accuracy in identifying the safety of empty mottled surfaces. The confusion matrix for these experiments as shown in Table 7 demonstrates that:

Precision (Safe) = 100%, recall (Safe) = 95.19%.

Precision (Unsafe) = 95.41%, recall (Unsafe) = 100%.

The proposed approach is able to assess the safety of the dropping area and recognize any change in the safety state during the entire dropping process under practical assumptions.

**Table 7.** Confusion matrix (experiments on recorded videos).

| | | Predicted Class | |
|---|---|---|---|
| | | **Safe** | **Unsafe** |
| Actual Class | Safe | 3855 | 195 |
| | Unsafe | 0 | 4050 |

*4.4. The Significance of the Proposed Approach*

In terms of the practical implications and feasibility of the proposed real-time safety classifier for the dropping zone, the experimental results indicate that the proposed approach provides reliable accuracy based on low-demand calculations. Additionally, one of the most essential aspects this approach introduces is scalability, which means it can function with a range of image/frame resolutions and dropping zone area sizes. It can also work if the drone is at various heights above the dropping zone.

In terms of the potential impact and benefits, the classifier assesses whether the area is suitable from a safety point of view. It also assesses the continuous safety of the zone during the dropping period—so as to prevent interruptions or accidents (e.g., a car or person entering the area) while the package is being dropped, resulting in potential cost savings for delivery companies. The speed of the classifier, operating at 30 frames per second, makes it suitable for the real-time checks required for the safety of a package-dropping exercise and provides an immediate warning for any possible danger that enters the dropping zone. This will assist delivery companies in ensuring that there is no accidental damage to the product during dropping or damage to either ground-based or drone-based property.

*4.5. Real-World Implementation Examples*

To demonstrate the effectiveness and practicality of the proposed approach, two case studies were implemented in a specific drone delivery scenario and videos were created as follows:

1.  A video was created to demonstrate how the suggested method classifies the dropping region in a fast and accurate manner (see "Assessment of drone delivery dropping zone" YouTube, uploaded by Assem Alsawy, 16 January 2019, https://www.youtube.com/watch?v=zC9TMOhJ-2g&ab_channel=EasyLearning (accessed on 3 January 2024)).

2.   Another video was created to demonstrate the proposed method's ability to segment and mask the territory beneath the drone into safe and unsafe areas as the drone navigates and hovers at different altitudes. (see "Safe/Unsafe areas masks for delivery drone dropping" YouTube, uploaded by Assem Alsawy, 16 January 2019, https://www.youtube.com/watch?v=nu_skGwiCUE&ab_channel=EasyLearning (accessed on 3 January 2024)).

*4.6. Challenges and Limitations*

The classifier has produced a fast and reliable path solution to dropping zone assessment of a predefined spot, and safety monitoring during dropping. However, there are several challenges that the proposed classifier faces:

- Since the proposed classifier is color-based, it does not provide a method that automatically recognizes the type of surface of the dropping zone.
- Since safety is not just about the dropping zone, the threat may come from outside this zone. There is a need for a method to examine the area around the dropping box and measure the risks that may occur.
- The proposed approach does not cover delivery on a moving platform such as ships, which would require a tracker that tracks the moving dropping area during the dropping process.
- The lack of safety in the dropping zone puts the dropping operation at risk of cancellation, which is costly for the delivery company. It is necessary to develop a method that finds another nearby and safe place to drop the package.
- Drones drop packages using a rope. We have not taken the rope into account in the assessment during the drop.
- GPS accuracy does not guarantee that the point the customer sets is the exact location the drone will go to. A visual approach is needed that integrates with GPS information for a more accurate and secure drop point.

## 5. Conclusions and Future Work

The proposed image-processing-based classifier is a simple and real-time classifier that assesses the safety of the dropping zone for a delivery drone before and during the dropping operation. This is the first classifier that aims to fully automate the delivery operations. Experiments on still images and videos of dropping zones show the classifier's high reliability using the precision-recall metric to recognize the safety status of empty spaces, whether it is a flat horizontal surface or a mottled surface, and when objects intrude into the dropping area. This approach considers the early phase operations of commercial delivery drone companies, where the dropping spot is pre-agreed with the customer prior to flight.

The classifier is limited in assessing the safety of vertical, sloping, elevated, and water surfaces. A more general classifier is needed to cover other types of surfaces that are not included in the proposed classifier, especially if/when the preclearance dropping spot assumption is removed. As machine learning methods have achieved high-precision results in various computer vision tasks, it is planned to apply deep learning techniques, along with spatial information, to produce a more sophisticated classifier.

For the drone to be able to interact with the environment with a semantic and spatial perception, it must be able to find another suitable spot to place the delivered package in cases where the primary dropping zone is not safe. The goal is to develop an approach that is able to find an alternative dropping zone through a two-step method: finding the empty boxes in the region of interest that can be used as candidate dropping sites, and ranking the candidate sites to choose the most appropriate one. Object detection may also play a role in a future solution, not every object will be considered a hazard in the process of dropping, so it will be necessary to develop an object recognition method to assess the type of risk according to the type of object identified.

**Author Contributions:** Conceptualization, A.A.; Methodology, A.A.; Software, A.A.; Validation, A.A.; Formal analysis, S.M.; Resources, A.H. and D.M.; Writing—original draft, A.A.; Writing—review & editing, A.A. and S.M; Supervision, S.M. All authors have read and agreed to the published version of the manuscript.

**Funding:** This project has received funding from the European Union's Horizon 2020 Research and innovation Programme under the Marie Skłodowska-Curie Cofunding of regional, national and international programmes Grant agreement No: 847402.

**Data Availability Statement:** Data are available upon request.

**Conflicts of Interest:** All authors declare that they have no conflicts of interest.

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
