# Peer review of "An Image Processing Approach for Real-Time Safety Assessment of Autonomous Drone Delivery"

_drones, doi:10.3390/drones8010021_

Round 1
Reviewer 1 Report
Comments and Suggestions for Authors
Summary/Contribution: The paper aims to study the similarities and differences in landing and delivery dropping processes of drones, as well as to build a real-time safety classifier for the dropping zone using an image segmentation method combined with a motion detector. The proposed approach in the paper focuses on evaluating the safety of drone delivery dropping zones and describes the evaluation metrics for assessing the safety of the dropping zone during the entire dropping process.
Comments/Suggestions:
- Provide a more detailed explanation of the proposed approach for evaluating the safety of drone delivery dropping zones, including the specific image segmentation method and motion detection techniques used.
- Include a comparative analysis of different existing approaches for drone landing automation in section 2, to provide a comprehensive overview of the current research landscape.
- Expand on the limitations and challenges of the clustering method used for object detection, particularly in misclassifying flat horizontal and mottled surfaces, and discuss potential solutions or alternative methods.
- Include a discussion on the practical implications and feasibility of the proposed real-time safety classifier for the dropping zone, considering factors such as computational requirements, real-world implementation challenges, and scalability.
- Provide more details on the evaluation metrics used to assess the safety of the dropping zone, including how precision and recall were calculated and any assumptions made in the experiments.
- Include a section on the dataset used for training and testing the proposed approach, including the size, diversity, and representativeness of the dataset, to ensure the generalizability of the results.
- Discuss the potential impact and benefits of the proposed approach, such as improved efficiency and safety in drone delivery operations, reduced risks of accidents or damage, and potential cost savings for delivery companies.
- Consider including a case study or real-world implementation example to demonstrate the effectiveness and practicality of the proposed approach in a specific drone delivery scenario.
- The authors need to add a short paragraph about the security aspects of drone communications.
- For this purpose, they may include the following references (and others) in their study: a. https://ieeexplore.ieee.org/document/9842403 b. https://www.mdpi.com/1424-8220/21/6/2057
Can be improved.
Author Response
Dear reviewer
We appreciate the time and effort that you have dedicated to providing your valuable feedback on our manuscript. It was your valuable and insightful comments that led to possible improvements in the current version. The authors have carefully considered the comments and tried our best to address them. We hope the manuscript after careful revisions meets your high standards. Here we provide a point-by-point response to your comments.
Comment 1: Provide a more detailed explanation of the proposed approach for evaluating the safety of drone delivery dropping zones, including the specific image segmentation method and motion detection techniques used.
Response: Thank you for this suggestion, more detailed explanations have been added to the proposed approach in Section 3.
Comment 2: Include a comparative analysis of different existing approaches for drone landing automation in section 2, to provide a comprehensive overview of the current research landscape.
Response: Thank you for pointing this out, a comparative analysis of different existing approaches has been added in section 2. A summary of this comparison is illustrated in this table
Paper |
Category |
Method |
Results/Evaluation |
Masataka Kan et al. (2018) |
Normal, non-visual landing |
Calibration of GPS (Global Positioning System)  |
The experimental drone was able to autonomously fly to the goal position. |
A Patrik et al. (2019) |
Normal, non-visual landing |
GNSS (Global Navigation Satellite System) |
The average deviation from the designated mark is less than one meter |
Y. Huang et al. (2017) |
Normal, visual landing |
Histogram of oriented gradients (HOG) and linear support vector machine algorithm (SVM) |
Overall Accuracy=84.0% |
Nguyen et al. (2017) |
Normal, visual landing |
Marker based-tracking algorithm |
Average error = 0.095%
|
MFR Lee et al. (2020) |
Normal, visual landing |
Fully convolutional neural network (FCNN) |
Qualitative results: There were some unsuccessful outcomes which were mainly due to imprecise motion control, and lag in the communication and processing. |
Nio et al. (2021)
|
Normal, visual landing |
Multi-scale QR codes for different heights |
The success rate is higher than 85% |
G Loureiro et al. (2019) |
Emergency, visual landing
|
Safe landing zone detection using light detection and ranging (LIDAR) data |
The detection algorithm had an average execution time of 37.87 ms. |
O. Bektash et al. (2020) |
Emergency, visual landing |
Tracking safe places to land, through a convolutional neural network. |
The framework has the potential to overcome uncertainties in operations with a safe profile and increased acceptance |
J.A. Gonzalez et al. (2021) |
Emergency, visual landing |
A lightweight convolutional neural network framework |
Real-time experiments validate the proposed strategy implemented in a low-cost mobile phone application and a commercial drone. |
G. Loureiro et al. (2021) |
Emergency, visual landing |
The analytical geometric characteristics + Principal Component Analysis (PCA) |
Evaluating the algorithm in simulated scenarios and an experimental dataset presenting suitability to be applied in real-time operations. |
Comment 3: Expand on the limitations and challenges of the clustering method used for object detection, particularly in misclassifying flat horizontal and mottled surfaces, and discuss potential solutions or alternative methods.
Response: We agree on this. We have added the following explanation in subsection 4.2, setting out the limitations and challenges of the clustering algorithm as set out by the reviewer we also, explained that the binary method under the industry assumptions is an alternative method. "The cluster-based classifier achieved complete accuracy in identifying the unsafeness of the dropping zones that contained objects and also provided high accuracy in identifying the unsafeness of vertical, sloping, and watery surfaces. On the other hand, it failed to recognize the safety of flat horizontal surfaces, mottled surfaces, and the unsafeness of the elevated surface. The reason behind that is the clustering method tried to extract the background of the image, assign one color to it, and assign other colors to different types of objects. Thus, it is suitable for detecting objects and vertical, sloped, and water surfaces. Flat horizontal surfaces and mottled surfaces are likely to be misclassified by this method, especially if the surface has paint or shadow. Elevated surfaces are recognized as a safe dropping zone with this method, but according to delivery companies, roofs are not suitable places to drop delivered packages. This method is not reliable in recognizing a safe dropping zone and it would have many false alarms to alert of unsafe when it is safe, but it scores a relatively good result in predicting unsafe zones."
Comment 4: Include a discussion on the practical implications and feasibility of the proposed real-time safety classifier for the dropping zone, considering factors such as computational requirements, real-world implementation challenges, and scalability.
Response: Thank you for pointing this out, a discussion has been added in subsection 4.4, to indicate the practical implications and feasibility of the proposed classifier.
Comment 5: Provide more details on the evaluation metrics used to assess the safety of the dropping zone, including how precision and recall were calculated and any assumptions made in the experiments.
Response: Thank you for this comment, More details on the evaluation metrics are provided in subsection 3.3, and some assumptions made in the experiments to meet the industry point of view are presented in subsection 4.2.1.
Comment 6: Include a section on the dataset used for training and testing the proposed approach, including the size, diversity, and representativeness of the dataset, to ensure the generalizability of the results.
Response: As suggested by the reviewer, a modification has been added to the dataset section 4.1, which shows there are two groups of image data used in this method to enable working on still images and a live camera as follows:
- The proposed solution aims first to examine the given dropping zone right before the dropping operation and show whether it is safe or unsafe to drop the delivery package. The dataset used to test this classifier consists of static images: 70 bounded boxes were selected as candidate dropping zones and cropped from 14 drone view images. These boxes represent diverse types of surfaces: 10 boxes of flat horizontal surfaces, 10 boxes of mottled surfaces, 10 boxes of vertical surfaces, 10 boxes of sloped surfaces, 10 boxes of elevated surfaces, 5 boxes of water surfaces, and 15 of these boxes contain objects such as cars, humans, trees, and pets. Figure 10 displays samples of the first group of the dataset.
- For tracking the safety of the dropping zone during the dropping process, 18 recorded-by-drones videos were used, each of which represents the dropping area during the dropping process. Each video has a frame rate of 15 frames/second and is 30 seconds long; the videos were recorded from different heights (12m, 14m, 16m, 18m) and using different side lengths of dropping area boxes (2m, 3m, 4m, 5m). Nine videos of them are for flat horizontal surfaces, the other nine are for surfaces mottled with irregular markings, shades, and colors. To simulate the changing of safety situations, various objects were entered at random times for separate 15 seconds into each video.
Comment 7: Discuss the potential impact and benefits of the proposed approach, such as improved efficiency and safety in drone delivery operations, reduced risks of accidents or damage, and potential cost savings for delivery companies.
Response: Thanks for your concern, We think this is an excellent suggestion. We have added a paragraph in subsection 4.4, that discusses the potential impact and benefits of the proposed approach and shows that the classifier assesses whether the area is suitable from a safety point of view. It also assesses the continuous safety of the zone during the dropping period - so as to prevent interruptions or accidents (e.g. car or person entering the area) while the package is being dropped, resulting in potential cost savings for delivery companies. The speed of the classifier, operating at 30 frames per second, makes it suitable for real-time checks required for the safety of a package-dropping exercise and provides an immediate warning for any possible danger that enters the dropping zone, This will assist delivery companies in ensuring that there is no accidental damage to the product during dropping or damage to either ground-based or drone-based property.
Comment 8: Consider including a case study or real-world implementation example to demonstrate the effectiveness and practicality of the proposed approach in a specific drone delivery scenario. 
Response: Thank you for pointing this out. We agree with this comment. Therefore, we included the following case studies:
- To demonstrate the effectiveness and practicality of the proposed approach, some real-world specific drone delivery scenario examples have been implemented, and a video was created to demonstrate how the suggested method accurately classifies the dropping region. (see “Assessment of drone delivery dropping zone” YouTube, uploaded by Assem Alsawy, 16 January 2019, “https://www.youtube.com/watch?v=zC9TMOhJ-2g&ab_channel=EasyLearning”)
- Another video was created to demonstrate the proposed method's ability to segment and mask the territory beneath the drone while navigating into safe and unsafe areas as the drone navigates and hovers at different altitudes. (see “Safe & Unsafe areas masks for delivery drone dropping” YouTube, uploaded by Assem Alsawy, 16 January 2019, “https://www.youtube.com/watch?v=nu_skGwiCUE&ab_channel=EasyLearning”) .
Comment 9: The authors need to add a short paragraph about the security aspects of drone communications. For this purpose, they may include the following references (and others) in their study: https://ieeexplore.ieee.org/document/9842403, and https://www.mdpi.com/1424-8220/21/6/2057
Response: We agree with this and have updated the introduction in section 1, with a paragraph discussing security considerations for drone communications.
Comments on the Quality of English Language: Can be improved.
Response: Every spelling and grammar mistake found has been Corrected.
Many thanks again, you are welcome to further constructive comments if any.
We look forward to hearing from you regarding our submission and to responding to any further questions and comments you may have.
Sincerely,
Authors
Reviewer 2 Report
Comments and Suggestions for Authors
The authors present an image processing-based classification approach for the safety verification of delivery drone dropping process at a predefined destination. The autonomous drone delivery is a very important and highly interesting topic and the proposed methodology has the potential to contribute to the safe and efficient implementation of autonomous delivery drone systems. However, the submitted manuscript appears to offer only minor enhancements compared to the previously published conference paper and it doesn’t reference that paper. It is essential that authors clearly state the specific improvements made to the current manuscript and highlight any novel ideas or findings that distinguish it from the conference paper.
Comments on the Quality of English Language
Minor revisions to the English language are necessary.
Author Response
Dear reviewer
We appreciate the time and effort that you have dedicated to providing your valuable feedback on our manuscript. It was your valuable and insightful comments that led to possible improvements in the current version. The authors have carefully considered the comments and tried our best to address them. We hope the manuscript after careful revisions meets your high standards.
Comment 1: The submitted manuscript appears to offer only minor enhancements compared to the previously published conference paper and it doesn’t reference that paper. It is essential that the authors clearly state the specific improvements made to the current manuscript and highlight any novel ideas or findings that distinguish it from the conference paper.
Response: Thank you for pointing this out. In response to this comment, a section has been added to the manuscript to highlight the improvements made and the reference is made to the conference paper. Also, we have added a paragraph that discusses the potential impact and benefits of the proposed approach and shows that it assesses whether the area is suitable from a safety point of view. It also assesses the continuous safety of the zone during the dropping period - so as to prevent interruptions or accidents (e.g. car or person entering the area) while the package is being dropped. The speed of the classifier, operating at 30 frames per second, makes it suitable for real-time checks required for the safety of a package-dropping exercise. This will assist delivery companies in ensuring that there is no accidental damage to the product during dropping or damage to either ground-based or drone-based property.
In comparison to the conference paper, this paper describes a more general and sophisticated approach for assessing the dropping area before and throughout the dropping process in terms of:
- Data preprocessing
- Using Gaussian blur instead of normal blur. In the case of images that were taken from a high altitude or in low light, the resulting image might have a lot of noise, Gaussian blur assists in minimizing this noise and softening the image, making a small object in the landing area more noticeable, and that helps to improve segmentation accuracy.
- Locate the dropping zone on the image and describe the area of the cropped square in a way that is independent of any output device, meaning it does not depend on camera characteristics (see equation 1, 2).
- Dataset
- In This paper, two types of datasets (still images and video frames) were tested, to meet the requirements of evaluating the dropping zone before and during the dropping process, compared to one type of dataset (video frames) that was used in the conference paper.
- To guarantee the generalizability of the conclusions, the datasets used are diverse and representative. i.e. Landforms, including flat horizontal surfaces, mottled surfaces, vertical surfaces, inclined surfaces, elevated surfaces, and water surfaces, are included in the dataset used to categorize still images that were captured right before the dropping operation.
- Proposed Solution/Results
- Two segmentation methods (binary segmentation and cluster segmentation) were used and compared.
- The proposed method not only classifies the dropping zone as safe or unsafe, but it also manages the whole dropping procedure. It recommends if the dropping operation should be continued or aborted. (See Fig. 8 for a flowchart)
- One of the most essential aspects we are introducing in current work is scalability, which means it can function with a range of image/frame resolutions and dropping zone area sizes, and also, can work if the drone is on various heights above the dropping zone.
- The proposed approach shows high accuracy in assessing the dropping zone using a still images dataset (see Table 5 and Figure 14).
- Case Study
- To demonstrate the effectiveness and practicality of the proposed approach, some real-world specific drone delivery scenario examples have been implemented, and a video was created to demonstrate how the suggested method accurately classifies the dropping region. (see “Assessment of drone delivery dropping zone” YouTube, uploaded by Assem Alsawy, 16 January 2019, “https://www.youtube.com/watch?v=zC9TMOhJ-2g&ab_channel=EasyLearning”)
- Another video was created to demonstrate the proposed method's ability to segment and mask the territory beneath the drone while navigating into safe and unsafe areas as the drone navigates and hovers at different altitudes. (see “Safe & Unsafe areas masks for delivery drone dropping” YouTube, uploaded by Assem Alsawy, 16 January 2019, “https://www.youtube.com/watch?v=nu_skGwiCUE&ab_channel=EasyLearning”).
Comment 2: Minor revisions to the English language are necessary.
Response: The manuscript was reviewed and improved for proper use of English.
Many thanks again, you are welcome to further constructive comments if any.
We look forward to hearing from you regarding our submission and to responding to any further questions and comments you may have.
Sincerely,
Authors
Round 2
Reviewer 1 Report
Comments and Suggestions for Authors
The authors considered my comments and suggestions. Good luck.
Comments on the Quality of English LanguageA final proofread would be useful.
Author Response
Dear reviewer
We appreciate the time and work you put into offering helpful feedback on our manuscript.
Comment 1: The authors considered my comments and suggestions. Good luck.
Response: Thank you very much, this enhancement was made possible by your useful and informative feedback.
Comments on the Quality of English Language: A final proofread would be useful.
Response: The manuscript was reviewed and improved for proper use of English.
Thank you again, We look forward to hearing from you regarding our new submission
Sincerely,
Authors
Reviewer 2 Report
Comments and Suggestions for Authors
The authors have made major improvements to the manuscript, clearly highlighting the improvements made compared to their previous conference paper. The revised manuscript explicitly points out specific improvements, novel ideas, and findings that distinguish it from previously published work.
Since the flowcharts in Figures 7 and 8 are nearly identical, with the only difference being in one step, I recommend merging these figures to eliminate redundancy and enhance clarity.
Consider changing the capitalization of 'The' to 'the' in different sections in the manuscript such as section 4.4 for grammatical consistency.
Address similar issues across different parts of the document.
Author Response
Dear reviewer
We appreciate the time and work you put into offering helpful feedback on our manuscript.
Comment 1: The authors have made major improvements to the manuscript, clearly highlighting the improvements made compared to their previous conference paper. The revised manuscript explicitly points out specific improvements, novel ideas, and findings that distinguish it from previously published work.
Response: Thank you very much, this enhancement was made possible by your useful and informative feedback.
Comment 2: Since the flowcharts in Figures 7 and 8 are nearly identical, with the only difference being in one step, I recommend merging these figures to eliminate redundancy and enhance clarity.
Response: Thank you for this suggestion, we agree with this and have provided only one flowchart, and pointed out the additional step in subsection 3.2.
Comments on the Quality of English Language: Consider changing the capitalization of 'The' to 'the' in different sections in the manuscript such as section 4.4 for grammatical consistency. Address similar issues across different parts of the document.
Response: Thank you for pointing this out, we've changed the capitalization in the new submission.
Thank you again, We look forward to hearing from you regarding our new submission
Sincerely,
Authors